# Insulin Resistance in Mitochondrial Diabetes

**DOI:** 10.3390/biom13010126

**Published:** 2023-01-07

**Authors:** Chika Takano, Erika Ogawa, Satoshi Hayakawa

**Affiliations:** 1Division of Microbiology, Department of Pathology and Microbiology, Nihon University School of Medicine, Tokyo 173-8610, Japan; 2Department of Pediatrics and Child Health, Nihon University School of Medicine, Tokyo 173-8610, Japan; 3Department of Pediatrics, Tokyo Metropolitan Hiroo Hospital, Tokyo 150-0013, Japan

**Keywords:** mitochondrial diabetes, insulin resistance, mitochondrial DNA mutation, transfer RNA modopathy

## Abstract

Mitochondrial diabetes (MD) is generally classified as a genetic defect of β-cells. The main pathophysiology is insulin secretion failure in pancreatic β-cells due to impaired mitochondrial ATP production. However, several reports have mentioned the presence of insulin resistance (IR) as a clinical feature of MD. As mitochondrial dysfunction is one of the important factors causing IR, we need to focus on IR as another pathophysiology of MD. In this special issue, we first briefly summarized the insulin signaling and molecular mechanisms of IR. Second, we overviewed currently confirmed pathogenic mitochondrial DNA (mtDNA) mutations from the MITOMAP database. The variants causing diabetes were mostly point mutations in the transfer RNA (tRNA) of the mitochondrial genome. Third, we focused on these variants leading to the recently described “tRNA modopathies” and reviewed the clinical features of patients with diabetes. Finally, we discussed the pathophysiology of MD caused by mtDNA mutations and explored the possible mechanism underlying the development of IR. This review should be beneficial to all clinicians involved in diagnostics and therapeutics related to diabetes and mitochondrial diseases.

## 1. Introduction

Insulin resistance (IR) is an inappropriate cellular response to insulin hormone in insulin-dependent cells [1,2]. IR is often described as the main characteristic of type 2 diabetes mellitus (DM). As the global escalation of type 2 diabetes necessitates urgent investigation, numerous studies of the molecular mechanism of IR have been reported [1,2,3,4]. Mitochondrial dysfunction is one of the crucial mechanisms underlying IR development. Since diabetes was reported to be caused by mitochondrial DNA (mtDNA) mutations in the 1990s [5], the relationship between diabetes and mitochondria has attracted attention [6]. The number of articles retrieved by searching “mitochondrial diabetes (MD)” in the PubMed database has been steadily increasing (Figure 1). Over 1000 articles were published in 2014, and 1807 were published in 2021. Multiple surveillance studies have been conducted to identify causative gene mutations among the diabetic population [7,8]. To date, numerous mtDNA mutations and their geological distribution have been reported (MITOMAP).

mtDNA mutations cause mitochondrial dysfunction, which sometimes subsequently develops into diabetes. However, what determines the development of diabetes in mitochondrial disease? There is a need to provide an overview of the pathogenic mutants and explore the common features of mtDNA mutations. Moreover, the American Diabetes Association classifies MD as “Other; genetic defects of the beta cell” [9]. The main pathology is insulin secretion failure in pancreatic β-cells due to impaired mitochondrial ATP production. Although accumulating evidence has shown that this mtDNA mutation causes insulin-dependent diabetes [10], MD due to the m.3243A>G mutation was originally found in a noninsulin-dependent diabetes (NIDDM) family [5]. It is still debated whether MD is characterized by only insulin secretion disability, such as in type 1 DM, by IR, such as in type 2 DM, or by a combination of both. As mitochondrial dysfunction is one of the important factors causing IR, it is also necessary to focus on IR as one of the pathophysiologies of MD.

In this special issue, we have outlined the pathophysiology of MD. First, we briefly summarized the mechanism of IR. Second, we overviewed currently confirmed pathogenic mtDNA mutations and selected the variants causing diabetes. Third, we reviewed their clinical features from previous reports. Finally, we discussed the pathophysiology of MD caused by mtDNA mutations and explored the possible mechanism underlying the development of IR. This perspective should contribute to a better understanding of diagnostics and therapeutics to care for patients with mitochondrial diseases.

## 2. Insulin Signaling and Molecular Mechanism of Insulin Resistance

Insulin signaling begins when insulin binds to the transmembrane insulin receptor [11]. Insulin activates a tyrosine kinase, which is followed by downstream events such as recruitment of the adaptor protein insulin receptor substrates (IRSs) [2]. Activated IRS-1 triggers subsequent signaling by binding to phosphoinositide 3-kinase (PI3K), which catalyzes the conversion of phosphatidylinositol 4,5-bisphosphate (PIP2) to phosphatidylinositol 3,4,5-trisphosphate (PIP3) [12]. PIP3 activates various protein kinases, including AKT, which facilitates glucose entry into cells by the translocation of GLUT-4 [13].

IR is a pathologic condition in which insulin-dependent cells, such as skeletal muscle and adipocytes, fail to respond properly to normal circulatory levels of insulin [1,2]. The possible underlying causes of IR have been well reviewed by Yaribeygi et al. [2]. In brief, high levels of plasma lipids due to overeating activate serine–threonine kinase, thereby inhibiting the insulin signaling pathway [14]. Obesity-induced inflammatory cytokines, particularly tumor necrosis factor-α (TNF-α), also impair insulin signaling via the serine phosphorylation of IRS-1. This effect reduces GLUT-4 expression, thus decreasing glucose entry into cells [15]. In addition to IRSs, several other substrates for mediating insulin action in an IRS-independent manner have been reviewed by de Luca et al. [14]. IR can occur through a decrease in insulin receptor affinity due to a mutation in the insulin receptor or antibodies against receptors [16,17]. Any point mutation in GLUT4 that alters this transporter could also impair glucose entry into cells and the downstream signaling pathways [18]. Obesity-induced endoplasmic reticulum stress affects insulin synthesis and disrupts proper insulin receptor synthesis, leading to impaired insulin signaling [19,20].

Mitochondrial dysfunction is another important mechanism of IR. Mitochondria are the main intracellular locations for generating adenosine triphosphate and free radical species, including reactive oxygen species (ROS) [21]. ROS enhance insulin sensitivity upon redox regulation of protein tyrosine phosphatase and the insulin receptor. However, chronic exposure to high ROS levels, in terms of an oxidative environment, could alter mitochondrial function and lead to the development of IR, β-cell dysfunction, and impaired glucose tolerance [22,23]. Moreover, the suppressed ability of the mitochondrial electron transport chain to oxidize NADH causes abnormally high concentrations of intramitochondrial and cytoplasmic NADH. The high NADH/NAD^+^ ratio could lead to ROS production and impaired β-oxidation, which are associated with IR in skeletal muscle [24,25].

## 3. Mitochondrial Diabetes among Confirmed Mitochondrial DNA Mutations

The mitochondrial genome comprises 16,569 nucleotides. This genome includes 13 protein-coding regions embedded in oxidative phosphorylation complexes I, III, IV, and V, two ribosomal RNAs (rRNAs) and 22 transfer RNAs (tRNAs). The two strands of the circular mtDNA chromosome have an asymmetric distribution. The displacement (D)-loop is a triple-stranded region generated by the synthesis of a short piece of H-strand DNA, 7S DNA (MITOMAP). Point mutations in the tRNA genes and the D-loop cause translation inhibition and mitochondrial genome instability, respectively [26].

Confirmed pathogenic mtDNA mutations are available from MITOMAP (accessed on 3 August 2022, https://www.mitomap.org/MITOMAP). To date, 80 types of mtDNA mutations have been confirmed as pathogenic mutations (Appendix A). The criteria of confirmation include the following: (1) independent reports of two or more unrelated families with evidence of similar disease; (2) evolutionary conservation of the nucleotide (for RNA variants) or amino acid (for coding variants); (3) presence of heteroplasmy; (4) correlation of variant with phenotype/segregation of the mutation with the disease within a family; (5) biochemical defects in complexes I, III, or IV in affected or multiple tissues; (6) functional studies showing differential defects segregating with the mutation (cybrid or single fiber studies); (7) histochemical evidence of a mitochondrial disorder; and (8) for fatal or severe phenotypes, the absence or extremely rare occurrence of the variant in large mtDNA sequence databases (MITOMAP). Fifty variants are tRNA mutations, and fewer than 30 variants are coding mutations. Among them, eight variants have been reported to demonstrate diabetes by searching previous findings (Table 1 and Figure 2).

Interestingly, almost all variants causing diabetes were single point mutations of the tRNA, except the m.9155A>G mutation. This mutation in the coding region of the ATP6 gene was first described as a novel mitochondrial diabetes and deafness (MIDD) in 2016 [43]. Although the detailed pathogenesis was not described, it is reasonable to suggest that this mutation causes diabetes because ATP6 contributes to proton-transporting ATP synthase activity. Healthy mitochondria in pancreatic β-cells produce ATP. The subsequent increase in the ATP/ADP ratio blocks ATP-sensitive K^+^ channels, resulting in the depolarization of the membrane potentials of β-cells. This step is followed by the influx of extracellular Ca^2+^ and insulin secretion. In contrast, impaired mitochondria lose their respiratory activity, causing a decrease in the influx of extracellular Ca^2+^ and insulin secretion. This evidence was first reported by Soejima et al. [10] and is supported by numerous studies. However, the pathophysiology by which point mutations in tRNA lead to diabetes is complex. Therefore, we focused on the variants leading to the recently described “tRNA modopathies”.

## 4. tRNA Modopathies

tRNA functions as an adaptor molecule that translates genetic information into amino acids. The 22 tRNAs in mitochondria translate essential subunits of the respiratory chain complexes. tRNA molecules are posttranscriptionally modified by nuclear-encoded tRNA modification enzymes [49]. Over 40 types of human tRNA modopathies play crucial roles in protein synthesis by regulating tRNA structure and stability and decoding genetic information on mRNA. Therefore, loss of tRNA modifications could result in severe pathological consequences, now referred to as “tRNA modopathies” [50,51]. Suzuki et al. revealed a complete picture of the posttranscriptional modification of human tRNAs encoded by mtDNA [49]. Accumulating evidence described below has elucidated the pathogenic mechanisms underlying tRNA modopathies.

### 4.1. m.3243A>G Mutation in tRNA ^Leu (UUR)^

The m.3243A>G mutation in the tRNA ^Leu (UUR)^ gene was first reported in patients with mitochondrial encephalomyopathy, lactic acidosis, and stroke-like episodes (MELAS) in Japan [27]. The mutation was highly frequent, observed in 26 of 31 independent MELAS patients. The pathogenicity of this mutation specifically causes the hypomodification of the 5-taurinomethyluridine U (τm^5^U) at position 34 modification of mt tRNA ^Leu (UUR)^ [28]. As the taurine modification stabilizes wobble anticodon-codon pairing, normal tRNA^Leu (UUR)^ efficiently pairs with codons UUA and UUG. In contrast, tRNA^Leu (UUR)^ with mtDNA mutation lacks the taurine modification, resulting in a specific reduction in UUG codon-specific translation but not UUA codon-specific translation. This translational aberration causes abnormal synthesis of mitochondrial-encoded proteins, resulting in tRNA modopathies [50]. In 1992, van den Ouweland et al. reported for the first time that this point mutation in mtDNA is a pathogenic factor for DM [5]. The m.3243A>G mutation was identified in a large pedigree with NIDDM, which was characterized by hyperglycemia and insulin resistance in combination with maternally inherited sensorineural hearing loss. This striking finding promoted research in this field. A Japanese surveillance study published in 1994 described the m.3243A>G mutation in patients with insulin-dependent diabetes mellitus (IDDM) and NIDDM [7]. Another study in 1994 described pancreatic β-cell secretory defects associated with mtDNA mutation of the tRNA ^Leu (UUR)^ gene, as insulin secretion investigated with insulinogenic index and urinary 24-h C-peptide immunoreactivity excretion was remarkably impaired in patients with the m.3243A>G mutation [29]. Some had normal glucose tolerance, while others had impaired glucose tolerance. Interestingly, plasma C-peptide immunoreactivity 6 min after glucagon injection was markedly reduced in patients with MD [29]. Another study also described similar clinical characteristics of an impaired insulin secretory response to oral glucose load [30]. Cutting-edge research using the pancreatic β-cell line MIN6 was reported by Soejima et al. in 1996 [10]. mtDNA depletion inhibited the glucose-stimulated increase in the intracellular free Ca^2+^ content as well as the elevation of insulin secretion. This evidence clarified the mechanism by which mtDNA depletion causes insulin-dependent DM. However, some previous reports mentioned the possible presence of IR in patients with the m.3243A>G mutation [31,32,33]. Although a German surveillance study in 2002 concluded that this mutation was unlikely causative of IR, the question of why some patients exhibited impaired insulin sensitivity could not be answered [34].

In 2002, Maassen summarized the clinical features of patients with the m.3243A>G mutation [35]. The patients develop diabetes around the age of 35. Although the mutation is present in patients with either type 1 or type 2 DM, patients with type 2 tend to rapidly develop insulin dependency. Most of these patients do not have a high body mass index. Impaired hearing due to reduced detection of high tone frequencies is present [35]. In 2003, Suzuki et al. also characterized patients with the m.3243A>G mutation using a cohort study conducted on 113 Japanese diabetic patients. The patients were short and thin in stature and showed an early middle-aged onset of diabetes and deafness. Most of the patients required insulin therapy due to progressive insulin secretory defects. Insulin secretory capacity was more severely impaired in patients whose mothers were glucose intolerant [36]. Interestingly, a recent study in Denmark conducted on nondiabetic carriers of the m.3243A>G mutation suggested that reduced insulin sensitivity could represent the earliest phase in pathogenesis [37].

### 4.2. m.3256 C>T, m.3260A>G, and m.3271T>C Mutations in tRNA ^Leu (UUR)^

As shown in Table 1, the point mutations m.3256C>T, m.3260A>G, and m.3271T>C share the same locus as m.3243A>G in tRNA ^Leu (UUR)^. Gerbitz et al. reviewed these mutations as different types of mtDNA mutations associated with diabetes in 1995 [38], but they did not describe m.3256C>T and m.3260A>G as causative mutations for diabetes. m.3256C>T causes myoclonus epilepsy associated with ragged red fibers (MERRF)-like syndrome, and m.3260A>G causes maternally inherited myopathy and cardiomyopathy. Tsukuda et al. investigated the prevalence and clinical characteristics of diabetes caused by tRNA ^Leu (UUR)^ mutations in 1997 [39]. They screened 440 diabetic patients with diabetic mothers for 11 mitochondrial gene mutations reported in mitochondrial neuromuscular disorders, including 3250, 3251, 3252, 3254, 3256, 3260, 3271, 3291, 3302, and 3303, in addition to an A to G transition at 3243. One subject carried m.3271T>C and seven carried m.3243A>G, while the rest of the patients screened had no mutations. The patient with the m.3271T>C mutation had excellent glycemic control with diet alone and had neither hearing impairment nor symptoms suggesting MELAS. Another report compared the clinical features of m.3271 T>C with m.3243 A>G and concluded m.3271 T>C as a mild type of MELAS syndrome [40]. In summary, although these variants share the locus of tRNA ^Leu (UUR)^, clinical features were not as distinct as those of the m.3243 A>G mutation, varied between individuals and were unlikely to be related to IR.

### 4.3. m.8344 A>G Mutation in tRNA ^Lys^ and m.14709 T>C Mutation in tRNA ^Glu^

The mutations m.8344A>G and m.14709T>C were found in the tRNA ^Lys^ and tRNA ^Glu^ genes, respectively. Healthy individuals have another taurine-containing modification, 5-taurinomethyl-2-thiouridine (τm^5^s^2^U) in tRNA^Lys^ and tRNA^Glu,^ in addition to τm^5^U in tRNA ^Leu (UUR)^ [52]. These also represent mitochondrial diseases caused by a deficiency in tRNA taurine modification at position 34. The lack of taurine modifications affects accurate mitochondrial translations of A- and G-ending codons [50]. tRNA^Lys^ with the m.8344A>G mutation is associated with MERRF. Hypomodified tRNA^Lys^ does not efficiently convert its cognate codons, impairing protein synthesis and leading to respiratory activity defects [49]. In 1994, Suzuki et al. first reported an association between this mutation and diabetes [41]. Insulin secretory capacity was significantly lower in these patients, suggesting pancreatic β-cell insulin secretion defects. A surveillance study showed that m.8344A>G was observed in 3 of 29 diabetic patients; two patients required insulin therapy at presentation [42]. The m.14709T>C mutation was reported in 1995 to be related to myopathy and diabetes mellitus. The patients showed impaired insulin secretion [45]. A patient with early onset was also reported, diagnosed with diabetes at age eight and requiring insulin administration at presentation [46]. This mutation was reported as MD-causative in Tunisian families [47] and, interestingly, was also reported in Italian families with NIDDM [48].

### 4.4. m.12258 C>A Mutation in tRNA ^Ser (AGY)^

The mutation m.12258C>A is located in tRNA ^Ser (AGY)^. tRNA ^Ser (AGY)^ lacks the entire D-loop [50], which has roles in mtDNA replication, nucleoid organization and nucleotide homeostasis [53]. In 1998, Lynn et al. first reported an association between this mutation and diabetes [44]. As the affected population size seems to be small, subsequent case reports were scarce; however, patients carrying m.12258C>A showed the highest likelihood of developing diabetes [42]. Two patients in this study did not require insulin treatment at presentation, whereas the incidence of progression to insulin requirement was 100%.

## 5. mtDNA Polymorphisms Associated with Insulin Resistance

As MITOMAP’s confirmed pathogenic mutations include rRNA/tRNA, coding and control region mutations, the m.16189T>C mutation in the D-loop region of mtDNA was not included. However, this mutation also attracts attention as a common mtDNA variant (mtDNA polymorphism, Figure 2). The m.16189T>C mutation was originally reported as a common variant [54] and was first described as a causative mutation of IR [55]. The prevalence of this variant significantly increased in individuals with progressively increasing fasting insulin concentrations. As this site of polymorphism is required for the regulation of mtDNA replication, it sometimes causes length variations in the mtDNA genome. Although there was still insufficient evidence to show the link between the polymorphism and IR [54,55], a surveillance study in China supported this clinical feature [56]. The prevalence of the m.16189T>C variant among type 2 diabetic patients with a maternal family history was significantly higher than that among the controls. The patients showed higher fasting serum insulin levels, decreased insulin sensitivity index, and higher HOMA-IR index [57,58].

## 6. Pathophysiology of Insulin Resistance in Mitochondrial Diabetes

In this special issue, we focused on mtDNA point mutations and organized the diabetes-causative mutations. Selected mtDNA mutations were point mutations in the tRNA gene and commonly tRNA modopathies. Moreover, almost all the tRNA modopathies were caused by an aberration in taurin modifications, which are present at position 34: τm^5^U in tRNA ^Leu (UUR)^ and τm^5^s^2^U in tRNA^Lys^ and tRNA^Glu^. These anticodon modifications directly regulate decoding, and mutations in the modification enzymes affect translation in a codon-specific manner [50]. In contrast, there have only been a few reports of point mutations causing diabetes in coding regions or mtDNA deletion. This finding suggests that codon-specific translational aberrations play a crucial role in MD development.

The main pathophysiology of MD is described as insulin secretion failure, as mitochondrial oxidative phosphorylation plays an important role in glucose-stimulated insulin secretion from pancreatic β-cells via ATP production [10]. When reviewing the clinical features of MD of tRNA modopathies, most patients require insulin treatment [35,36]. However, some patients certainly revealed low insulin sensitivity and glucose intolerance. The following three points describe the possible mechanisms involved in IR caused by MD.

(1) As described in the section on insulin signaling and the molecular mechanism of insulin resistance, disruption of the mitochondrial electron transport chain causes an increase in the NADH/NAD^+^ ratio. This effect reduces muscle oxidative capacity and impairs β-oxidation, subsequently leading to the development of skeletal muscle IR [24,25]. Importantly, the lack of the mt tRNA^Leu (UUR)^ tm^5^U34 modification specifically affects the translation of ND6 mRNA because of the enrichment of the UUG codon in the mRNAs [50,59]. As ND6 is a component of complex I of the electron transport chain, impaired expression of ND6 causes instability of the mitochondrial electron transport chain and subsequent accumulation of NADH, possibly resulting in peripheral IR. This hypothesis could explain the presence of reduced insulin sensitivity in noninsulin-dependent m.3243A>G carriers observed in a study in Denmark [37]. The diversity of clinical features could be explained by the heteroplasmy of mtDNA mutations in pancreatic β-cells as well as in skeletal muscles.

(2) mtDNA alterations affect the efficiency of oxidative phosphorylation, which consequently leads to reduced membrane potential and ATP synthesis and increased ROS production [26]. Therefore, all mitochondrial diseases caused by mtDNA mutations have the potential to develop into IR, such as that seen in type 2 DM. Peterson suggested that an age-associated decline in mitochondrial function contributes to insulin resistance in the elderly [60]. A recent study suggested age-corrected heteroplasmy as a prognostic marker in carriers of the m.3243A>G mutation [61], although there is insufficient evidence to show that the accumulation of heteroplasmy with age causes progressive MD.

(3) mtDNA mutations sometimes overlap in a single individual. The coexistence of mtDNA polymorphisms might be associated with IR. As a cytoplasmic tRNA modopathy, CDKAL1-mediated thiomethylation of cytoplasmic tRNA^Lys^
_UUU_ at position 37 also attracts attention as a causative gene for type 2 DM [62,63]. More recently, through the International Diabetes Federation projects, an Asian-specific mtDNA variation, m.1382A>C, was found to lead to a K14Q amino acid replacement in MOTS-c (mitochondrial open reading frame of the 12S rRNA-c), which is an insulin-sensitizing mitochondrial-derived peptide. Of interest, this finding contributes to the risk management of type 2 DM [64]. A comprehensive study exploring the population structure of mtDNA was reported from England [65]. Such a biobank referencing system will enable the identification of novel mtDNA-phenotype associations. In addition, while not discussed in detail in this special issue, numerous pathogenic nuclear-coding genes associated with mitochondrial diseases have been discovered [66]. Further investigations are required to elucidate the complex interactions between mitochondria and the nuclear genome and explain the diversity of mitochondrial diseases.

Accumulated evidence enables us to discuss the possible mechanisms underlying IR development. The first hypothesis is the most plausible and likely provides a clear understanding of the relationship between tRNA modopathy and IR. However, why taurine modifications specifically lead to the development of diabetes remains an open question. From the clinician’s perspective, we should be aware that patients may have early impairment of glucose metabolism at the prediabetic stage. It might be beneficial to monitor glucose intolerance among patients with tRNA modopathies.

## 7. Diagnostics and Therapeutics for Mitochondrial Diabetes

Japan has been at the forefront of mitochondrial disease in diagnostics [7,28], clinical characteristics [36], and treatment [52,67] since the 1990s. Recent contributions to this field, especially in pediatrics, have been reported by us and others [68,69]. As we have described in this review, mitochondrial disease may cause diabetes via insulin secretory defects and/or IR. Therefore, the presence of mtDNA mutations and related genetic defects should be considered to achieve better management for any type of diabetes. Although the aforementioned clinical features of MD [36] help to achieve the appropriate diagnosis, investigations should not be limited within these clinical features because of the diversity of phenotypes. The urinary epithelium has been recommended as a diagnostic sample to detect mitochondrial heteroplasmy [70].

Recently, effective therapeutics for mitochondrial diseases have been developed. Taurine supplementation as tRNA modopathy-targeted medicine has had a striking impact on this field. In 2002, Suzuki et al. reported that taurine conjugates with mitochondrial tRNA^Leu (UUR)^ or tRNA^Lys (UUU)^ for proper codon–anticodon interaction to facilitate the synthesis of mitochondrial-encoded proteins [52]. In clinical trials, oral taurine supplementation in MELAS patients showed a remarkable therapeutic effect in preventing stroke-like episodes by increasing modification of taurine-conjugated uridine of mitochondrial tRNA^Leu (UUR)^ [71,72]. Homma et al. suggested that taurine improved mitochondrial function in iPSCs generated from a MELAS patient [73]. The recent clarification of posttranscriptional modifications of tRNAs encoded in mtDNA [49] might accelerate studies on therapeutics targeting tRNA modifications. In parallel with the new approach, coenzyme Q10 [74], l-carnitine [75], and vitamin cocktails, widely used for mitochondrial diseases, are all noninvasive treatments. As the treatment might also be effective for preventing the development of skeletal muscle IR and MD caused by tRNA modopathy, we must not deprive patients of the opportunity to receive treatment. When the involvement of mitochondrial disease is suspected, the use of these treatments should be considered with prompt and appropriate diagnostics.

## 8. Poor Prognosis of COVID-19 Patients Complicated with DM and Mitochondria: A Possible Missing Link

Coronavirus disease 2019 (COVID-19) has been a significant health concern worldwide since its first appearance in 2019. Diabetes mellitus and advanced age are major risk factors for the prognosis of COVID-19-infected individuals. However, the mechanisms are poorly understood. We propose the hypothesis that mitochondrial dysfunction in immune cells and the entire body of patients with these complications is a possible reason for this poor prognosis. Infection with coronaviruses causes endoplasmic reticulum stress in cells, which disrupts protein folding and leads to apoptosis. Mitochondria are involved in this process, and open reading frame 9b of severe acute respiratory syndrome coronavirus 2 (SARS-CoV-2) acts on mitochondria to suppress the type I interferon response in infected cells; this also induces autophagy. However, apoptosis and autophagy are conversely suppressed in cells persistently infected with the virus [76]. This is why infected cells remain in vivo for a long time and continue to produce viruses. Diabetic patients and elderly individuals both demonstrate mitochondrial dysfunction and may be more susceptible to hijacking by SARS-CoV-2 [77]. Analysis of mitochondrial mutations and dysfunction in patients with poor prognoses may yield new therapeutic strategies.

## 9. Closing Remarks

Numerous observational pieces of evidence support the symbiosis hypothesis that mitochondria are descended from aerobic bacteria that parasitized anaerobic bacteria in prehistoric times. Over billions of years, mitochondria have lost their uniqueness as independent organisms. However, they still have their own genome, mutate stochastically, and are subject to selection pressure from the environment separately from the host genome. In particular, recent academic interest has focused on interactions with the microbiota, including the gastrointestinal flora [78,79,80]. tRNA mutations have been reported in several bacteria, many of which can be explained by interactions with other bacteria in the flora [81]. In other words, mutations may be favored in special environments. The frequent occurrence of tRNA modopathies in DM, as we have described in this review, may have some evolutionary explanation. Diabetes is found only in multicellular organisms with circulatory systems, i.e., vertebrates. Before that, various variants in cellular glucose metabolism must have arisen and disappeared as an adaptation to the environment. It is possible that some of these variants were preserved in evolution or were newly mutated during evolution and spread by matrilineal inheritance. If they are lethal, of course, they will not remain; thus, there may be a trade-off allowing their persistence, such as reduced susceptibility to infection or adaptation to starvation.

## Figures and Tables

**Figure 1 biomolecules-13-00126-f001:**
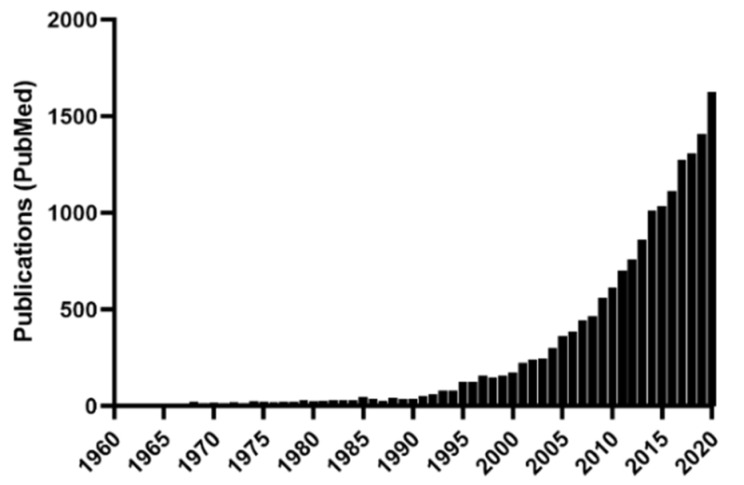
The number of articles retrieved by searching “mitochondrial diabetes” in the PubMed database.

**Figure 2 biomolecules-13-00126-f002:**
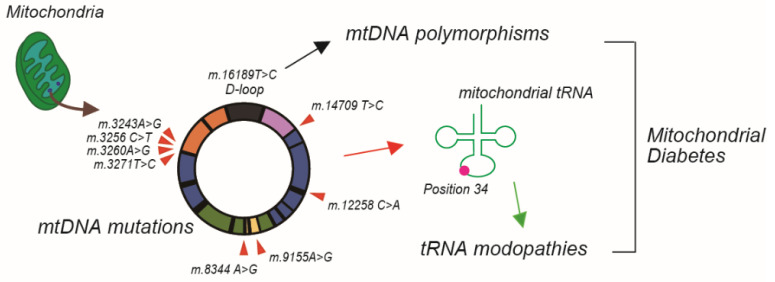
Mitochondrial diabetes-related mitochondrial DNA mutations.

**Table 1 biomolecules-13-00126-t001:** Point mutations of mitochondrial DNA associated with diabetes.

	Locus Type	Locus	Associated Diseases	Allele	Position	aaΔ or RNA	Reference
1	tRNA	MT-TL1	MELAS/Leigh Syndrome/DMDF/MIDD/SNHL/CPEO/MM/FSGS/ASD/Cardiac+multi-organ dysfunction	m.3243A>G	3243	tRNA Leu (UUR)	[5,7,10,27,28,29,30,31,32,33,34,35,36,37]
2	tRNA	MT-TL1	MELAS; possible atherosclerosis risk	m.3256C>T	3256	tRNA Leu (UUR)	[38,39]
3	tRNA	MT-TL1	MMC/MELAS	m.3260A>G	3260	tRNA Leu (UUR)	[38,39]
4	tRNA	MT-TL1	MELAS/DM	m.3271T>C	3271	tRNA Leu (UUR)	[38,39,40]
5	tRNA	MT-TK	MERRF; Other—LD/Depressive mood disorder/leukoencephalopathy/HiCM	m.8344A>G	8344	tRNA Lys	[41,42]
6	Coding	MT-ATP6	MIDD, renal insufficiency	m.9155A>G	9155	Q210R	[43]
7	tRNA	MT-TS2	DMDF/RP + SNHL	m.12258C>A	12258	tRNA Ser (AGY)	[42,44]
8	tRNA	MT-TE	MM + DMDF/Encephalomyopathy/Dementia + diabetes + ophthalmoplegia	m.14709T>C	14709	tRNA Glu	[45,46,47,48]

MELAS: Mitochondrial Encephalomyopathy, Lactic Acidosis, and Stroke-like episodes, DMDF: Diabetes Mellitus + DeaFness, MIDD: Mitochondrial Diabetes and Deafness, SNHL: SensoriNeural Hearing Loss, CPEO: Chronic Progressive External Ophthalmoplegia, MM: Mitochondrial Myopathy, FSGS: Focal Segmental Glomerular Sclerosis, ASD: Autism Spectrum Disorder, MMC: Maternal Myopathy and Cardiomyopathy, DM: Diabetes Mellitus, MERRF: Myoclonic Epilepsy and Ragged Red Muscle Fibers, LD: Leigh Disease, RP: Retinitis Pigmentosa

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
