# Peer review of "Insulin Resistance in Mitochondrial Diabetes"

_biomolecules, 2023, doi:10.3390/biom13010126_

Round 1

Reviewer 1 Report

The manuscript is a Review article summarizing knowledge on mitochondrial diabetes and insulin resistance as one clinical feature of it. The manuscript is nicely written and well-structured. 

Specific points and suggestions.

1. Figure one could be expanded to show, in addition top the current panel, a period 1960-2000 with a better resolution, because it is hard to estimate the count of papers during this period in the current figure. Moreover, there can be some numbers added to the Figure, Figure legend, or both, to illustrate, for example, the count of articles every 5 years. Overall, some improvements can be made to the Figure to make it more informative by increasing the description in the text and a count-year resolution.

2. Kindly check Table 1. It would be more useful for readers to have this Table in a Text format, not as a current Figure view. The are also many arrows on the Table, which probably appear due to copying the file from the text document by using a screenshot option.

3. Lines 227-234 are in bold. Although it will be fixed during English editing and proofreading, kindly check that the text is consistent, all spaces and dots are in place throughout the manuscript, and kindly fix small typos and grammar where needed.

4. Line 265 ends with "as follows;". Kindly check if this is the best way to end the paragraph.

5. The manuscript only includes one Figure and one Table. More illustrations would be beneficial to have.

Author Response

The authors would like to thank the reviewer for the careful and thorough review of this manuscript and for the thoughtful comments and constructive suggestions. These comments and suggestions helped to improve the quality of this manuscript. The author’s responses are shown below (the reviewer’s comments are in blue/italics).

Reviewer 1

  1. Figure one could be expanded to show, in addition top the current panel, a period 1960-2000 with a better resolution, because it is hard to estimate the count of papers during this period in the current figure. Moreover, there can be some numbers added to the Figure, Figure legend, or both, to illustrate, for example, the count of articles every 5 years. Overall, some improvements can be made to the Figure to make it more informative by increasing the description in the text and a count-year resolution.

Thank you for your suggestion. We modified Figure 1 and added the following sentences:

L.33-35

The number of articles retrieved by searching “mitochondrial diabetes (MD)” in the PubMed database has been steadily increasing (Figure 1). Over 1,000 articles were published in 2014, and 1,807 were published in 2021.

  1. Kindly check Table 1. It would be more useful for readers to have this Table in a Text format, not as a current Figure view. The are also many arrows on the Table, which probably appear due to copying the file from the text document by using a screenshot option.

We organized Table 1 according to the reviewer’s suggestion.

  1. Lines 227-234 are in bold. Although it will be fixed during English editing and proofreading, kindly check that the text is consistent, all spaces and dots are in place throughout the manuscript, and kindly fix small typos and grammar where needed.

Thank you for noticing an inappropriate format. We confirmed these sentences and fixed the format.

  1. Line 265 ends with "as follows;". Kindly check if this is the best way to end the paragraph.

We modified the sentence.

Line 271-272  The following three points describe possible mechanisms involved in IR caused by MD.

  1. The manuscript only includes one Figure and one Table. More illustrations would be beneficial to have.

Thank you for the constructive suggestion. We added one more illustration as Figure 2.

Reviewer 2 Report

The manuscript is devoted to highly demanded and relevant scientific problem as the theme of the association of mitochondrial dysfunction with diabetes cause growing concern in academical society nowadays. 

About 60% of references were published over a 10 years ago. For the past 5 years a number of systhematic reviews and meta-analysis devoted to mitohondrial DNA mutations and diabetes were published. The last decade many articles on the field of mDNA studiing (for example, an Atlas of mDNA genotype-phenotype associations, UK Biobank, 2021) were relised which also may be mentioned in the paper.

Author Response

The authors would like to thank the reviewer for the careful and thorough review of this manuscript and for the thoughtful comments and constructive suggestions. These comments and suggestions helped to improve the quality of this manuscript. The author’s responses are shown below (the reviewer’s comments are in blue/italics).

About 60% of references were published over a 10 years ago. For the past 5 years a number of systhematic reviews and meta-analysis devoted to mitohondrial DNA mutations and diabetes were published. The last decade many articles on the field of mDNA studiing (for example, an Atlas of mDNA genotype-phenotype associations, UK Biobank, 2021) were relised which also may be mentioned in the paper.

Thank you for the constructive comment. We added recent articles as references [6, 65, and 75], including the article suggested by the reviewer, and modified the sentences as follows:

Line 301-304    A comprehensive study exploring the population structure of mtDNA was reported from England [65]. Such a biobank referencing system will enable the identification of novel mtDNA-phenotype associations.